# Conformal Prediction as Bayesian Quadrature

**Jake C. Snell** [1]    **Thomas L. Griffiths** [1] [2]

## Abstract

As machine learning-based prediction systems are increasingly used in high-stakes situations, it is important to understand how such predictive models will perform upon deployment. Distribution-free uncertainty quantification techniques such as conformal prediction provide guarantees about the loss black-box models will incur even when the details of the models are hidden. However, such methods are based on frequentist probability, which unduly limits their applicability. We revisit the central aspects of conformal prediction from a Bayesian perspective and thereby illuminate the shortcomings of frequentist guarantees. We propose a practical alternative based on Bayesian quadrature that provides interpretable guarantees and offers a richer representation of the likely range of losses to be observed at test time.

## 1. Introduction

Machine learning systems based on deep learning are increasingly used in high-stakes settings, such as medical diagnosis or financial applications. These settings impose unique constraints on the performance of these systems: we want them to produce good outcomes in the aggregate, but also do so fairly and with a guarantee of a low probability of harm. However, predictive models based on deep learning can be difficult to interpret, and commercial models increasingly tend to offer little information about the techniques used in training. This creates a new challenge: How can we flexibly and reliably quantify the suitability of a model for deployment without making too many assumptions about how the model was trained or in which settings it will be used?

Recent research on quantifying uncertainty has employed methods based on conformal prediction (Vovk et al., 2005),

which aim to provide guarantees for model performance in a distribution-free way. However, these techniques are based on ideas from frequentist statistics, making it difficult to incorporate prior knowledge that might be available about specific models. For example, in a particular setting we might have access to some information about the distribution of the data that is likely to be encountered, and can construct tighter guarantees on the performance of models by making use of this information. Moreover, they focus on controlling the expected loss averaged over many unobserved datasets rather than focusing on the actual set of observations.

In this paper, we show how methods for guaranteeing model performance can be understood and extended by viewing them from a Bayesian perspective. We develop a framework in which we explicitly model uncertainty in the quantile values associated with particular observations, providing a nonparametric tool for characterizing possible distributions where the model might be deployed that is appropriately constrained by observed data. This framework allows us to draw upon methods from the fields of statistical prediction analysis (Aitchison & Dunsmore, 1975) and probabilistic numerics (Cockayne et al., 2019; Hennig et al., 2022) to develop guarantees that are interpretable and make adaptive use of available information.

We show that two popular uncertainty quantification methods, split conformal prediction (Vovk et al., 2005; Papadopoulos et al., 2002) and conformal risk control (Angelopoulos et al., 2024), can both be recovered as special cases of our framework. Our approach gives a more complete characterization of the performance of these approaches, as we are able to determine the full distribution of possible outcomes rather than a single point estimate. Since our approach is grounded in Bayesian probability, we can easily incorporate knowledge relevant to evaluating the performance of these models when it is present, such as monotonicity or distributional assumptions, while defaulting to existing methods when absent. Our results show that Bayesian probability, while it is often discarded due to the apparent need to specify prior distributions, is actually well-suited for distribution-free uncertainty quantification.

---

[1]Department of Computer Science, Princeton University [2]Department of Psychology, Princeton University. Correspondence to: Jake C. Snell <jsnell@princeton.edu>.

*Proceedings of the 42nd International Conference on Machine Learning*, Vancouver, Canada. PMLR 267, 2025. Copyright 2025 by the author(s).

## 2. Background

Conformal prediction methods apply a wrapper on top of black-box predictive models to be able to subject them to statistical analysis. In order to generate meaningful predictions about future performance, it is assumed that we have access to a small calibration dataset that is representative of the deployment conditions. Performance on this dataset then provides the foundation for generating predictions about future performance. We begin by reviewing existing current distribution-free uncertainty quantification techniques and Bayesian quadrature methods.

### 2.1. Distribution-free Uncertainty Quantification Techniques

Uncertainty quantification techniques provide guarantees on the future performance of a black-box predictive model mapping inputs $X$ to outputs $Y$ based on a calibration set consisting of $X_1, \ldots, X_n$ and $Y_1, \ldots, Y_n$. Different approaches do so in different ways. For more information on these techniques, refer to Shafer & Vovk (2008) or Angelopoulos & Bates (2023).

**Split Conformal Prediction**  The goal of Split Conformal Prediction (Vovk et al., 2005; Papadopoulos et al., 2002) is to generate a prediction set or interval that contains the ground-truth output with high probability. This is often expressed in terms of the coverage level $1 - \alpha$. It relies on a score function $s(x, y)$ which measures the disagreement between a predictor's output and the ground truth.

The conformal guarantee is

$$\Pr\left(Y_{n+1} \notin \mathcal{C}(X_{n+1})\right) \leq \alpha, \qquad (1)$$

where

$$\mathcal{C}(X_{n+1}) = \{y : s(X_{n+1}, y) \leq \hat{q}\} \qquad (2)$$

and $\hat{q}$ is the $\frac{\lceil(n+1)(1-\alpha)\rceil}{n}$ quantile of $s_1 = s(X_1, Y_1), \ldots, s_n = s(X_n, Y_n)$. Here, $\mathcal{C}(X_{n+1})$ is a prediction set or interval which aims to include the ground-truth output.

**Conformal Risk Control**  In Conformal Risk Control (Angelopoulos et al., 2024), the goal is to generalize conformal prediction to more general loss functions that are monotonic functions of a single parameter $\lambda$. Conformal Risk Control (CRC) proceeds by viewing the coverage guarantee (1) as the expected value of a 0-1 loss. It is assumed that the maximum possible value of the loss is $B$ and that the problem is "achievable" by design in that there exists some setting $\lambda_{\max}$ that satisfies the conformal guarantee. Additionally, each loss function $L_i(\lambda)$ is assumed to be a monotonic non-increasing function of $\lambda$. The guarantee offered by Conformal Risk Control is of the form

$$E\left(\ell(\mathcal{C}_{\hat{\lambda}}(X_{n+1}), Y_{n+1})\right) \leq \alpha, \qquad (3)$$

where

$$\hat{\lambda} = \inf\left\{\lambda : \frac{n}{n+1}\hat{R}_n(\lambda) + \frac{B}{n+1} \leq \alpha\right\} \qquad (4)$$

and $\hat{R}_n(\lambda) = \frac{1}{n}\sum_{i=1}^{n} L_i(\lambda)$ is the empirical risk.

### 2.2. Bayesian Quadrature

Bayesian quadrature (Diaconis, 1988; O'Hagan, 1991) is a general technique for evaluating integrals that allows for uncertainty in the integrand. It estimates the value of an integral $\int_a^b f(x)\,dx$ by the following four steps: (1) place a prior $p(f)$ on functions, (2) evaluate $f$ at $x_1, x_2, \ldots, x_n$, (3) compute a posterior given the observed values of $f$ by Bayes' rule, and (4) estimate $\int_a^b f(x)\,dx$. Suppose that $f(x_i) = y_i$ for $i = 1, 2, \ldots, n$. The posterior over $f$ is

$$p(f \mid x_{1:n}, y_{1:n}) \propto p(f)\prod_{i=1}^{n}\delta(y_i - f(x_i)), \qquad (5)$$

where $\delta(\cdot)$ is the Dirac delta function. The posterior mean then provides an estimate for the integral:

$$\int_a^b f(x)\,dx \approx \int_a^b f_n(x)\,dx, \text{ where} \qquad (6)$$

$$f_n(t) = E(f(t) \mid x_{1:n}, y_{1:n}). \qquad (7)$$

It has been demonstrated that many classical quadrature procedures such as the trapezoid rule can be recovered by placing a Gaussian process prior on functions (Karvonen & Särkkä, 2017).

### 2.3. Summary and Prospectus

Bayesian quadrature provides an illustration of how a primarily numerical method can be connected to Bayesian inference, and in doing so potentially admit additional information about the underlying function that can be incorporated via a prior distribution. In next section, we will see how a similar approach can be applied to conformal prediction, identifying a Bayesian framework that reproduces existing distribution-free uncertainty quantification techniques. The challenge in doing so is that we want guarantees of the style obtained from Bayesian models, but we want to make the approach as general as possible in its assumptions about the underlying distribution. We solve this problem via an approach inspired by probabilistic numerics to construct a nonparametric characterization of the underlying distribution based on the calibration set.

## 3. Decision-theoretic Formulation

In this section we show how split conformal prediction and conformal risk control can be formulated as instances of a general decision problem.

Let $z = (z_1, \ldots, z_n)$ be a set of calibration data where each observation $z_i = (x_i, y_i)$ consists of an input and a ground truth label. Let $\theta$ denote the true state of nature that defines a shared density $f(z_i \mid \theta)$ for the data.[1] A new test point $z_{\text{new}}$ is assumed to have the same distribution. Let $\lambda$ be a control parameter (e.g. threshold) that must be chosen based on the calibration data. We assume the presence of a loss function $L(\theta, \lambda)$ which quantifies the loss incurred by selecting $\lambda$ when the true state of nature is $\theta$.

The decision-theoretic goal is to choose a decision rule $\lambda(z)$ that controls the *risk*:

$$R(\theta, \lambda) = \int L(\theta, \lambda(z)) f(z \mid \theta) \, dz. \quad (8)$$

It is often desirable to choose $\lambda$ so that it is robust to any possible state of nature $\theta$. The *maximum risk* is defined as

$$\bar{R}(\lambda) = \sup_{\theta} R(\theta, \lambda). \quad (9)$$

In distribution-free uncertainty quantification applications, it is often trivial to achieve arbitrarily low risk (for example by forming prediction sets covering the entire output space). We thus want to find decision rules whose risk is upper bounded by a constant $\alpha$:

$$\bar{R}(\lambda) \leq \alpha, \quad (10)$$

and use another criterion (such as expected prediction set size) to select among these. We call a rule that satisfies (10) an $\alpha$-*acceptable decision rule*.

### 3.1. Recovering Split Conformal Prediction

We now show how split conformal prediction is a special case of this decision-theoretic problem. Let $L_{\text{scp}}(\theta, \lambda)$ be the *miscoverage loss*:

$$\begin{aligned} L_{\text{scp}}(\theta, \lambda) &= \Pr\{s(z_{\text{new}}) > \lambda\} \quad (11) \\ &= 1 - \Pr\{s(z_{\text{new}}) \leq \lambda\} \\ &= 1 - \int \mathbb{1}\{s(z_{\text{new}}) \leq \lambda\} f(z_{\text{new}} \mid \theta) \, dz_{\text{new}}, \end{aligned}$$

where $s$ is an arbitrary nonconformity function.

**Proposition 3.1.** *Define $s_i \triangleq s(z_i)$ for $i = 1, \ldots, n$ and let $s_{(1)} \leq s_{(2)} \leq \ldots \leq s_{(n)}$ be the corresponding order*

---

[1]In the interest of notational convenience, we assume densities and integrals over $z_i$ but these may be replaced by probability mass functions and summations as appropriate.

*statistics. Let $\lambda_{\text{scp}}$ be the following decision rule:*

$$\lambda_{\text{scp}} = \begin{cases} s_{(\lceil (n+1)(1-\alpha) \rceil)}, & \text{if } \lceil (n+1)(1-\alpha) \rceil \leq n \\ \infty, & \text{otherwise.} \end{cases} \quad (12)$$

*Then $\lambda_{\text{scp}}$ is an $\alpha$-acceptable decision rule for the miscoverage loss $L_{\text{scp}}$ defined in (11).*

*Proof.* Proofs for all theoretical results may be found in Appendix B. □

Therefore the prediction set can be constructed as in (2):

$$\mathcal{C}_{\text{scp}}(x_{\text{new}}) = \{y \in \mathcal{Y} : s(x_{\text{new}}, y) \leq \lambda_{\text{scp}}\}, \quad (13)$$

and by Proposition 3.1, $\mathcal{C}_{\text{scp}}$ satisfies the conformal guarantee from (1).

### 3.2. Recovering Conformal Risk Control

Conformal risk control generalizes split conformal prediction by considering losses that are monotonic non-increasing functions of a single parameter $\lambda$.

$$L_{\text{crc}}(\theta, \lambda) = \int \ell(z_{\text{new}}, \lambda) f(z_{\text{new}} \mid \theta) \, dz_{\text{new}}, \quad (14)$$

where $\ell(z_{\text{new}}, \lambda)$ is an individual loss function that is monotonically non-increasing in $\lambda$.

**Proposition 3.2.** *Let $\lambda_{\text{crc}}$ be the following decision rule:*

$$\lambda_{\text{crc}} = \inf \left\{ \lambda : \frac{1}{n+1} \left( \sum_{i=1}^{n} \ell(z_i, \lambda) + B \right) \leq \alpha \right\}. \quad (15)$$

*Then $\lambda_{\text{crc}}$ is an $\alpha$-acceptable decision rule for $L_{\text{crc}}$ defined in (14).*

Note in particular that when $\ell(z, \lambda)$ can be expressed in the form $\ell(\mathcal{C}_\lambda(x_{n+1}), y_{n+1})$, this recovers the conformal risk control guarantee from (3).

## 4. Our Approach

We introduce our approach by reinterpreting split conformal prediction and conformal risk control as special cases of a more general Bayesian procedure. In order to do so, we borrow ideas from both Bayesian quadrature (Diaconis, 1988; O'Hagan, 1991) and distribution-free tolerance regions (Guttman, 1970). Bayesian quadrature (Section 2.2) solves a numerical integration problem by placing a prior on functions and using Bayesian inference to compute a distribution over the value of the integral. Distribution-free tolerance regions provide a distribution over quantile spacings that holds regardless of the original underlying distribution. Putting these ideas together allows us to extend

conformal prediction by producing bounds on expected loss tailored to the actual losses observed in the calibration set.

The remainder of this section is structured as follows. In Section 4.1, we discuss the relationship between risk control and Bayes risk. In Section 4.2, we describe a general approach for using Bayesian quadrature to bound the posterior risk. In Section 4.3, we make the quadrature "distribution-free" by removing the dependence on a prior over functions. In Section 4.4 we handle uncertainty in the evaluation locations of the function by applying results that characterize the spacing between consecutive quantiles. In Section 4.5, we show how to use these results to produce an upper bound on the expected loss. Finally, in Section 4.6, we show how previous conformal prediction techniques can be viewed as a special case of our procedure that only considers the expectation of the posterior loss.

### 4.1. Bayes Risk

The risk $R(\theta, \lambda)$ measures the expected loss for one who already knows the true state of nature $\theta$ but not the particular data observed. However, in practical applications the situation is reversed: we *do* know the observed data but there is uncertainty about the state of nature. Therefore, we want a decision rule that protects against high loss for a range of possible $\theta$. This idea is expressed as the *integrated risk*:

$$r(\pi, \lambda) = \int R(\theta, \lambda)\pi(\theta)\, d\theta, \qquad (16)$$

where the prior $\pi(\theta) \geq 0$ measures the relative importance of the different possible states of nature. It is well-known that the minimizer of the integrated risk is the so-called *Bayes decision rule*:

$$\lambda^{\pi} \triangleq \underset{\lambda}{\operatorname{argmin}}\, r(\lambda \mid z), \qquad (17)$$

where $r(\lambda \mid z)$ is the *posterior risk*

$$r(\lambda \mid z) = E(L_{\lambda} \mid z) = \int L(\theta, \lambda(z))\pi(\theta \mid z)\, d\theta, \quad (18)$$

and $\pi(\theta \mid z) \propto \pi(\theta) f(z \mid \theta)$. Interestingly, the worst-case integrated risk of a decision rule is identical to its maximum risk (9)

$$\bar{r}(\lambda) \triangleq \sup_{\pi} r(\pi, \lambda) = \sup_{\theta} R(\theta, \lambda) = \bar{R}(\lambda). \qquad (19)$$

We can therefore focus on bounding the worst-case integrated risk $\bar{r}(\lambda)$, since this will also bound the maximum risk $\bar{R}(\lambda)$.

### 4.2. Reformulation as Bayesian Quadrature

We now turn our attention to finding $\lambda$ minimizing the posterior risk (18). Consider risks that can be expressed as the expectation over individual losses:

$$L(\theta, \lambda) = \int \ell(z_{\text{new}}, \lambda) f(z_{\text{new}} \mid \theta)\, dz_{\text{new}}. \qquad (20)$$

It is well-known that the expectation of a random variable is equal to the definite integral of its quantile function over its domain (Shorack, 2000, p. 116). Consider the distribution function of individual losses induced by $\lambda$ for a particular value of $\theta$:

$$F(\ell) \triangleq \Pr\{\ell(z_{\text{new}}, \lambda) \leq \ell \mid \theta\} \qquad (21)$$

The corresponding quantile function is:

$$K(t) \equiv F^{-1}(t) = \inf\{\ell : F(\ell) \geq t\}, \qquad (22)$$

and the expected loss given $K$ is simply $\int_0^1 K(t)\, dt$.

Instead of performing posterior inference over $\theta$, we propose to take an approach inspired by Bayesian quadrature that places a corresponding prior over $K$. Figure 1 shows a schematic overview of Bayesian quadrature in this setting and how our proposed approach differs. The posterior risk given the observed individual losses $\ell_i \triangleq \ell(z_i, \lambda)$ for $i = 1, \ldots, n$ becomes:

$$E(L \mid \ell_{1:n}) = \int J[K] p(K \mid \ell_{1:n})\, dK, \qquad (23)$$

where $J[K] \triangleq \int_0^1 K(t)\, dt$ and we have suppressed the dependence on $\lambda$ for notational convenience. The posterior over quantile functions can be expressed as:

$$p(K \mid \ell_{1:n}) = \int p(K \mid t_{1:n}, \ell_{1:n}) p(t_{1:n} \mid \ell_{1:n})\, dt_{1:n} \tag{24}$$

$$p(K \mid t_{1:n}, \ell_{1:n}) \propto \pi(K) \prod_{i=1}^{n} \delta(\ell_i - K(t_i)). \qquad (25)$$

This resembles the Bayesian quadrature problem from Section 2.2, except the evaluation sites $t_1, \ldots, t_n$ are unknown. Fortunately, the distribution of $t_1, \ldots, t_n$ is independent of the true distribution of the losses, as we shall now show.

### 4.3. Elimination of the Prior Distribution

In order to address the dependence of the posterior risk on the prior $\pi(K)$, we derive an upper bound on the posterior expected loss. The bound takes the form of a weighted sum of the observed losses, where the weights are determined by the spacing between consecutive quantiles.

**Theorem 4.1.** *Let $t_{(0)} = 0$, $t_{(n+1)} = 1$, and $\ell_{(n+1)} = B$. Then*

$$\sup_{\pi} E(L \mid t_{1:n}, \ell_{1:n}) \leq \sum_{i=1}^{n+1} u_i \ell_{(i)}, \qquad (26)$$

*where $u_i = t_{(i)} - t_{(i-1)}$.*

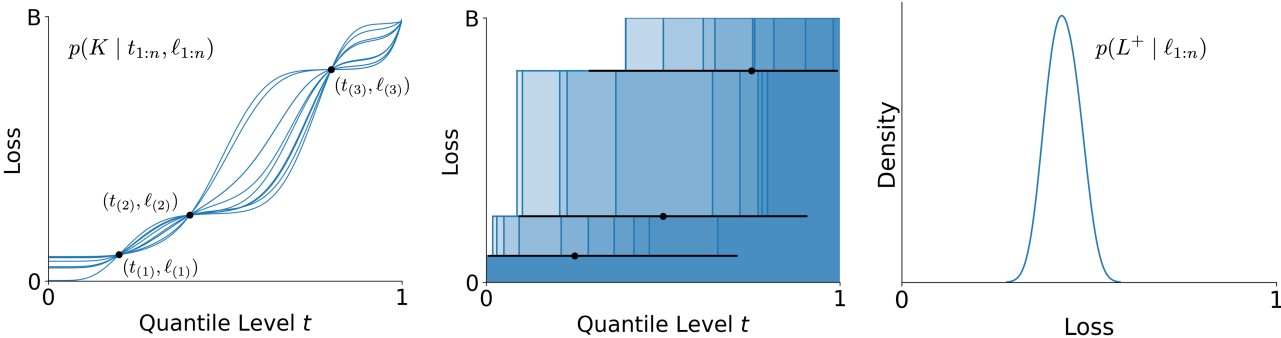

*Figure 1.* Overview of our approach. Left: Standard Bayesian quadrature places a prior over the quantile function of the loss distribution. The posterior is formed via Bayes' rule after observing a set of loss values and quantile levels. However, in practice quantile levels are not directly observed. Middle: Our approach combines properties of quantile spacings with a right rectangular integration rule to construct an upper bound on the posterior distribution of the expected loss. Randomly sampled spacings and corresponding quantile functions are shown in blue along with a 95% credible interval for each quantile level in black. Right: The posterior distribution for a random variable $L^+$ that upper bounds the expected loss is constructed by integrating over the unknown quantile levels.

Theorem 4.1 is based on the definite integral of the "worst-case" quantile function that is consistent with the observations. This strategy eliminates the need to specify a prior or evaluate an integral over functions $K$. We now turn our attention to handling the uncertainty over the quantiles $t_{1:n}$.

### 4.4. Random Quantile Spacings

We now appeal to a result about distribution-free tolerance regions that characterizes the distribution of spacings between consecutive ordered quantiles. Knowledge of this distribution will allow us to handle the input noise in the quadrature problem.

**Lemma 4.2** (Distribution of Quantile Spacings (Aitchison & Dunsmore, 1975, p. 140)). *Suppose that $\ell_1, \ldots, \ell_n$ are drawn i.i.d. with continuous[2] distribution function $F$. Let $t_i = F(\ell_i)$ and $u_i = t_{(i)} - t_{(i-1)}$, where by convention $t_{(0)} = 0$ and $t_{(n+1)} = 1$. Then $(u_1, u_2, \ldots, u_{n+1}) \cong \mathrm{Dir}(1, \ldots, 1)$.*

We are now ready to present our algorithm for bounding the expected loss $E(L \mid \ell_{1:n})$.

### 4.5. Bound on Maximum Posterior Risk

Putting together Lemma 4.2 and Theorem 4.1 allows us to bound the maximum posterior risk.

**Theorem 4.3.** *Define $\ell_{(i)}$ to be the order statistics of $\ell_1, \ldots, \ell_n$ for $i = 1, \ldots, n$ and $\ell_{(n+1)} \triangleq B$. Let $L^+$ be*

*the random variable defined as follows:*

$$U_1, \ldots, U_{n+1} \sim \mathrm{Dir}(1, \ldots, 1), \quad L^+ = \sum_{i=1}^{n+1} U_i \ell_{(i)}. \quad (27)$$

*Then for any $b \in (-\infty, B]$,*

$$\inf_\pi \Pr(L \le b \mid \ell_{1:n}) \ge \Pr(L^+ \le b). \quad (28)$$

Theorem 4.3 states that $L^+$ stochastically dominates the posterior risk[3], which allows us to directly form upper confidence bounds as follows.

**Corollary 4.4.** *For any desired confidence level $\beta \in (0, 1)$, define*

$$b_\beta^* = \inf_b \{ b : \Pr(L^+ \le b \mid \ell_{1:n}) \ge \beta \}. \quad (29)$$

*Then $\inf_\pi \Pr(L \le b \mid \ell_{1:n}) \ge \beta$ for any $b \ge b_\beta^*$.*

The critical value $b_\beta^*$ can be calculated by applying techniques for bounding linear combinations of Dirichlet random variables (Ng et al., 2011, p. 63). Alternatively, straightforward Monte Carlo simulation of $L^+$ is often sufficient, and is the approach we take in our experiments. An illustration is shown in Figure 2.

### 4.6. Recovering Conformal Methods

This perspective puts the previous distribution-free uncertainty techniques in a new light. Taking the expected value

---

[2]The correspondence to a Dirichlet distribution holds exactly for continuous distributions. Weighted sums of Dirichlet random variates stochastically dominate weighted sums of discrete quantile spacings, and thus due to space constraints we only consider continuous distributions here.

[3]We note that the distribution of the $U_i$ conditions just on the order of the losses and not their specific values. This assumption is consistent with the frequentist treatment of conformal prediction, which assumes that a new observation is equally likely to fall into each of the intervals formed by the loss values and the endpoints.

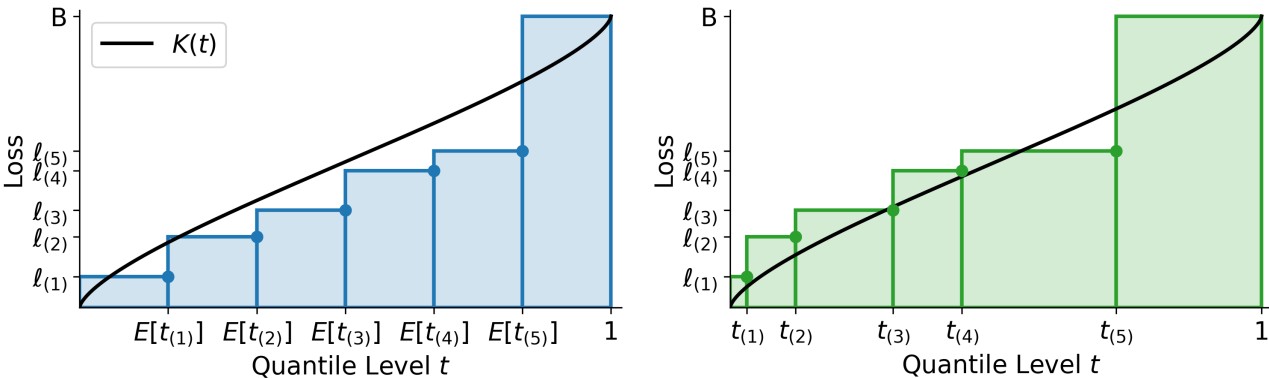

*Figure 2.* Our Bayesian approach to conformal prediction accounts for the variability in quantile levels better than previous approaches. Left: Conformal Risk Control (Angelopoulos et al., 2024) considers only the expectation over the unobserved quantile values $t_1, \ldots, t_n$. This can underestimate the true expected loss (shown here: estimated expected loss 0.45 vs. true expected loss 0.50). Right: Our approach makes use of the fact that the quantile spacings are drawn from a Dirichlet distribution. By considering the full distribution over quantiles, we gain a more complete view of the expected loss. Shown here is one sample drawn from this distribution, which estimates the expected loss as 0.58.

of $L^+$, we find

$$E(L^+) = \sum_{i=1}^{n+1} E(U_i)\ell_{(i)} = \frac{1}{n+1}\left(\sum_{i=1}^{n} \ell_i + B\right). \quad (30)$$

The Conformal Risk Control decision rule (15) then is simply the infimum over $\lambda$ for which $E(L^+) \leq \alpha$.

For, split conformal prediction, the individual loss is defined as $\ell_i = 1 - \mathbb{1}\{s_i \leq \lambda\}$. Therefore, suppose that $\lambda = s_{(k)}$. The expected value of $L^+$ then becomes:

$$E(L^+) = \frac{1}{n+1}\left(n + 1 - \sum_{i=1}^{n} \mathbb{1}\{s_i \leq s_{(k)}\}\right) \quad (31)$$

$$= 1 - \frac{k}{n+1} \quad (32)$$

Therefore, $E(L^+) \leq \alpha$ is satisfied whenever $k \geq (n+1)(1-\alpha)$, and in particular by $k^* = \lceil(n+1)(1-\alpha)\rceil$. This recovers (12) when $\lceil(n+1)(1-\alpha)\rceil \leq n$.

Putting these results together, we have recovered standard conformal prediction techniques but have the additional flexibility of considering the distribution of $L^+$ rather than the expected value alone. Our experiments explore the value of this approach.

## 5. Experiments

The primary goal of our experiments is to demonstrate the utility of producing a posterior distribution over the expected loss. We conduct experiments on both synthetic data and calibration data collected from MS-COCO (Lin et al., 2014). For each data setting, we randomly generate $M = 10,000$ data splits. Each method is used to select $\lambda$ with the goal

of controlling the risk such that $R(\theta, \lambda) \leq \alpha$ for unknown $\theta$. We compare algorithms on the basis of both the relative frequency of incurring risk greater than $\alpha$ and the prediction set size of the chosen $\lambda$. The ideal algorithm would select $\lambda$ such that the relative frequency of exceeding the target risk is at most a target failure rate of $1 - \beta = 0.05$ while minimizing prediction set size.

As demonstrated in Section 4.6, our method recovers conformal risk control by taking the expected value of $L^+$. Therefore, in order to demonstrate the effect of targeting a conditional guarantee (as opposed to a marginal one as in conformal risk control), we use our Bayesian quadrature-based method to compute the decision rule based on the one-sided highest posterior density (HPD) interval:

$$\lambda_{\text{hpd}}^{\beta} \triangleq \inf_{\lambda}\{\lambda : \Pr(L^+ \leq \alpha \mid \ell_{1:n}) \geq \beta\}, \quad (33)$$

by finding the corresponding critical values $b_\beta^*$ according to (29) via Monte Carlo simulation of Dirichlet random variates with 1000 samples. We include Risk-controlling Prediction Sets (RCPS) (Bates et al., 2021) with Hoeffding upper confidence bound as an additional baseline. Code for our experiments is publicly available on Github.[4]

### 5.1. Synthetic Binomial Data

We first sample directly from a known loss distribution so that we can directly compute the frequency of excessively large risk. Here the loss distribution is chosen to be a scaled binomial distribution, normalized to have a maximum loss of $B = 1$ and probability of failure set to $1 - \lambda$. This was

---

[4]https://github.com/jakesnell/
conformal-as-bayes-quad

Table 1. Relative frequency of trials (out of 10,000) for which the resulting decision rule $\lambda$ exceeded the target risk threshold $\alpha$.

| Decision Rule | Relative Freq. | 95% CI |
|---|---|---|
| CRC | 21.20% | [20.40%, 22.01%] |
| RCPS | 0.00% | [0.00%, 0.04%] |
| Ours ($\beta = 0.95$) | 0.03% | [0.01%, 0.09%] |

Note: Error bars are computed as 95% Clopper-Pearson confidence intervals for binomial proportions.

simulated by computing

$$\ell(z_i, \lambda) = \frac{1}{K} \sum_{k=1}^{K} \mathbb{1}\{V_{ik} > \lambda\}, \qquad (34)$$

where $V_{ik} \sim \text{Uniform}(0, 1)$ for $i = 1, \ldots, n$ and $k = 1, \ldots, K$. This loss is therefore monotonically non-increasing in $\lambda$ and achieves zero loss at $\lambda_{\max} = 1$. We set $n = 10$, $K = 4$, and $\alpha = 0.4$.

Since the expectation of the loss (34) is $1 - \lambda$, any trial for which $\lambda < 0.6$ constitutes a risk exceeding the $\alpha$ threshold. The relative frequency of trials exceeding this risk threshold are tabulated in Table 5.1. A histogram of the chosen $\lambda$ for each of the methods across all 10,000 trials is shown in Figure 3. For conformal risk control, the mean risk across all trials was $0.3363 \pm 0.0007$ and for our approach $\lambda_{\text{hpd}}^{0.95}$ the mean risk was $0.1758 \pm 0.0006$. In order to visualize the distribution of $L^+$, we plot a histogram of $L^+$ according to (27) estimated with 100,000 Dirichlet samples for three settings of $\lambda \in \{0.7, 0.8, 0.9\}$. The results are shown in Figure 4.

### 5.2. Synthetic Heteroskedastic Data

In this experiment we also use 10,000 random trials. We use $n = 200$ calibration samples each. To achieve heteroskedasticity, we let $X \sim U[0, 4]$ and $Y \mid X \sim \mathcal{N}(0, X^2)$. The prediction intervals are then formed as $[-\hat{\lambda}, \hat{\lambda}]$ where $\hat{\lambda}$ is selected by each method. The loss is the miscoverage loss and the target loss is set to $\alpha = 0.1$ (i.e. 90% coverage). The maximum allowable risk failure rate is set to 5% (i.e. $\beta = 0.95$). The results are show in Table 5.2.

### 5.3. False Negative Rate on MS-COCO

We also compare methods on controlling the false negative rate of multilabel classification on the MS-COCO dataset (Lin et al., 2014). The experimental setup mirrors that used by Angelopoulos & Bates (2023, Section 5.1). Each random split contains 1000 calibration examples and 3952 test examples. The results of this experiment are summarized in Table 5.3.

## 6. Discussion

Our results in Table 5.1 demonstrate that even though the Conformal Risk Control marginal guarantee holds, a significant number of individual trials (21.20%) may incur risk exceeding the target threshold. In contrast, by using the more conservative HPD criterion, very few of the trials (0.03%) exceeded the target risk. In Table 5.2, both RCPS and our method achieve failure rate below the target of 5% but our method achieves significantly smaller prediction intervals.

These results point to the qualitative difference in a marginal guarantee, which averages over many possible yet unobserved data sets vs. a conditional guarantee which focuses on knowledge about the state of nature conditioned on the calibration data actually observed. Previous work on conditional guarantees (Barber et al., 2021; Gibbs et al., 2024) has focused on input-conditional guarantees, where the guarantee is conditioned on for all in the input domain. Guarantees of this nature have been shown to be generally impossible without stronger distribution assumptions. Our guarantees are perhaps better characterized by the term "data-conditional guarantee", where we condition on the set of observed loss values. Our experiments demonstrate the practical benefits of this by achieving decisions that produce smaller prediction sets and intervals while not violating the constraint on maximum allowable failure rate. Our guarantees, in contrast, do not rely on strong distribution assumptions that would be necessary to produce an input-conditional guarantee.

The results are again confirmed in Table 5.3 on MS-COCO, which show that the marginal guarantees of Conformal Risk Control lead to an even greater percentage of trials exceeding the risk threshold. On the other hand, RCPS is able to control the risk but this comes at the cost of larger prediction sets. Our approach successfully balances these two concerns, producing prediction intervals that are shorter than baselines while not exceeding the maximum acceptable failure rate. It is also clear that the distribution of the expected loss upper bound $L^+$ in Figure 4 provides a more complete view of the range of possible losses and its dependence on $\lambda$, a perspective that is not offered by previous methods.

Our goal in this work is to show that the Bayesian viewpoint unlocks a richer interpretation compared to previous works, which focus on marginal guarantees that as we have shown in the paper correspond to the posterior mean. In order to draw an explicit correspondence between our work and previous approaches, the dependence on the prior was removed in Section 4.3. The intuition is that that any rational decision maker operating according to the rules of probability, regardless of prior (sufficiently expressive), would agree with the upper-bounding distribution of we derive. Naturally, commitment to a specific choice of prior would

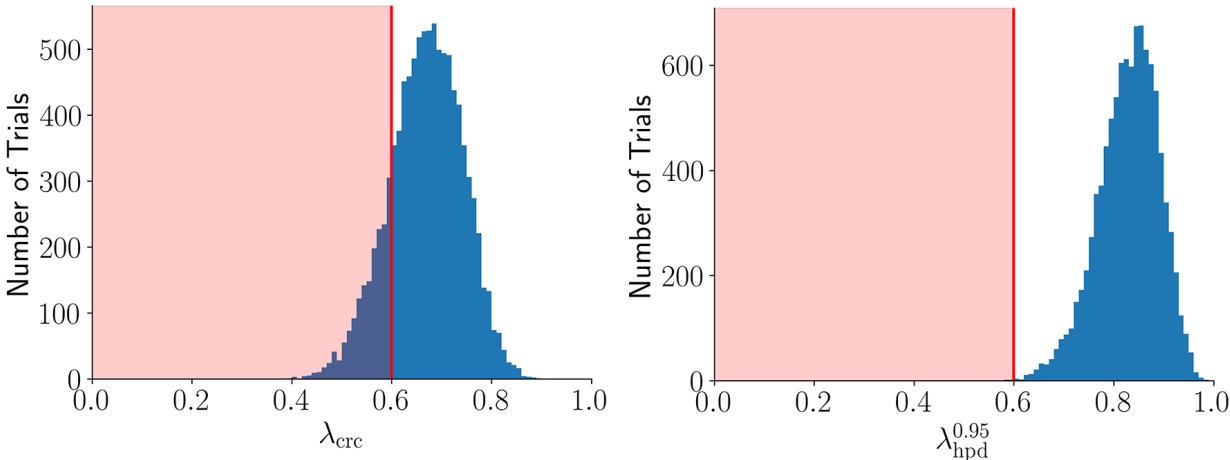

*Figure 3.* Comparison of risk incurred by each procedure across multiple trials. Left: Histogram of the decision rule $\lambda_{\text{crc}}$ chosen by Conformal Risk Control across $M = 10{,}000$ randomly sampled calibration sets. The region where per-trial risk exceeds $\alpha$ is highlighted in red. Right: Histogram of the $\lambda_{\text{hpd}}^{0.95}$ chosen according to our 95% Bayesian posterior interval.

*Table 2.* Relative frequency of trials (out of 10,000) for which the resulting decision rule $\lambda$ exceeded the target risk threshold $\alpha$ in the synthetic heteroskedastic experiment.

| Decision Rule | Relative Freq. | 95% CI | Mean Prediction Interval Length |
|---|---|---|---|
| Split Conformal Prediction / CRC | 46.19% | [45.21%, 47.17%] | 7.99 |
| RCPS | 0.0% | [0.0%, 0.04%] | 14.29 |
| Ours ($\beta = 0.95$) | 3.42% | [3.07%, 3.80%] | 9.50 |

Note: Error bars are computed as 95% Clopper-Pearson confidence intervals for binomial proportions.

*Table 3.* Results on MS-COCO comparing relative frequency of trials for which the resulting decision rule $\lambda$ exceeded the target risk threshold $\alpha$ and average prediction set size.

| Method | Relative Freq. | Pred. Set Size |
|---|---|---|
| CRC | 45.05% | 2.92 |
| RCPS | 0.0% | 3.57 |
| Ours ($\beta = 0.95$) | 5.43% | 3.04 |

lead to tighter distributions over the posterior risk, and in future work we seek to bridge these fields even further by exploring specific choices of priors over quantile functions.

The limitations of our method lie primarily in the two main assumptions it makes. First, it assumes that the data at deployment time are independent and identically distributed to the calibration data. Second, it assumes an upper bound $B$ on the losses. If either of these assumptions do not hold, then the guarantees produced by our method are no longer valid. Additionally, the bounds produced by our method are conservative in the sense that they hold for any choice of prior for the loss distribution (provided that the prior is consistent with the calibration data). Therefore, if the two

aforementioned assumptions do hold, the actual loss values may be significantly less than indicated by our method.

Overall, our approach demonstrates how conformal prediction techniques can be recovered and extended using Bayesian probability, all without having to specify a prior distribution. This Bayesian formulation is highly flexible due to its nonparametric nature, yet is amenable to incorporating specific information about the distribution of losses likely to be encountered. In practical applications, maximizing the risk with respect to all possible priors may be too conservative, and thus future work may explore the effect of specific priors on the risk estimate.

## 7. Related Work

**Statistical Prediction Analysis.** Statistical prediction analysis (Aitchison & Dunsmore, 1975) deals with the use of statistical inference to reason about the likely outcomes of future prediction tasks given past ones. Within statistical prediction analysis, the area of distribution-free prediction assumes that the parameters or the form of the distributions involved cannot be identified. This idea can be traced back to Wilks (1941), who constructed a method to form

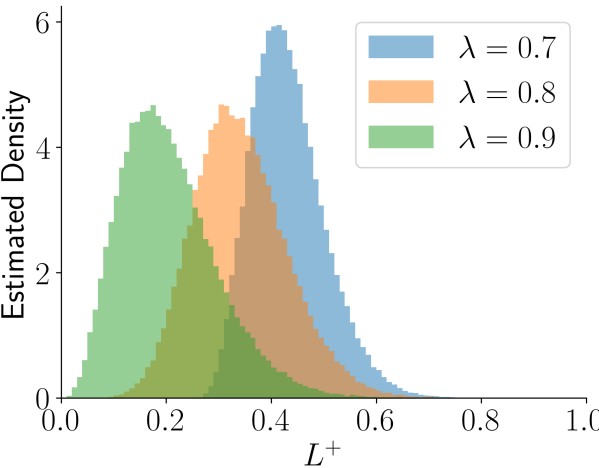

*Figure 4.* Probability density for $L^+$ with $\lambda \in \{0.7, 0.8, 0.9\}$ estimated using 100,000 Dirichlet samples.

distribution-free tolerance regions. Tukey (1947; 1948) generalized distribution-free tolerance regions and introduced the concept of statistically equivalent blocks, which are analogous to the intervals between consecutive order statistics of the losses. Much of the relevant theory is summarized by Guttman (1970), and the Dirichlet distribution of quantile spacing is discussed by Aitchison & Dunsmore (1975). We build upon these works by connecting them to Bayesian quadrature and applying them in the more modern context of distribution-free uncertainty quantification.

**Bayesian Quadrature.** The use of Bayesian probability to represent the outcome of a arbitrary computation is termed *probabilistic numerics* (Cockayne et al., 2019; Hennig et al., 2022). Since our approach is fundamentally based on integration, we focus primarily on the relationship with the more narrow approach of Bayesian quadrature, which employs Bayes rule to estimate the value of an integral. A lucid overview of this approach is discussed under the term *Bayesian numerical analysis* by Diaconis (1988), who traces it back to the late nineteenth century (Poincaré, 1896). The use of Gaussian processes in performing Bayesian quadrature is discussed in detail by O'Hagan (1991). Our approach is formulated similarly but differs in two main ways: (a) we use a conservative bound instead of an explicit prior, and (b) we have input noise induced by the random quantile spacings.

**Distribution-Free Uncertainty Quantification.** Relevant background on distribution-free uncertainty quantification techniques is discussed in Section 2.1. A recent and comprehensive introduction to conformal prediction and related techniques may be found in (Angelopoulos & Bates, 2023). Some recent works, like ours, also make use of quantile functions (Snell et al., 2023; Farzaneh et al., 2024) but re-

main grounded in frequentist probability. Another class of related work seeks to endow conformal prediction with properties naturally enjoyed by Bayesian methods, including the ability to incorporate prior information (Hoff, 2023), produce predictive distributions (Vovk et al., 2017), or adapt to time-varying data distributions (Prinster et al., 2024). Stanton et al. (2023) show how to apply conformal prediction to Bayesian optimization, another setting where maintaining a posterior over functions is important. Separately, Bayesian approaches to predictive uncertainty are popular (Hobbhahn et al., 2022) but make extensive assumptions about the form of the underlying predictive model. To our knowledge, we are the first to apply statistical prediction analysis and Bayesian quadrature in order to analyze the performance of black-box predictive models in a distribution-free way.

## 8. Conclusion

Safely deploying black-box predictive models, such as those based on deep neural networks, requires developing methods that provide guarantees of their performance. Existing techniques for solving this problem are based on frequentist statistics, and are thus difficult to extend to incorporate knowledge about the situation in which models may be deployed. In this work we provided a Bayesian alternative to distribution-free uncertainty quantification, showing that two popular existing methods are special cases of this approach. Our results show that Bayesian probability can be used to extend uncertainty quantification techniques, making their underlying assumptions more explicit, allowing incorporation of additional knowledge, and providing a more intuitive foundation for constructing performance guarantees that avoid overly-optimistic guarantees that can be produced by existing methods.

## Impact Statement

This paper introduces a practical algorithm for computing a posterior distribution for the expected loss based on the observed losses from a set of calibration data. The intended purpose of this algorithm is for the posterior distribution to inform deployment decisions of black-box predictive systems (e.g. deep neural networks) in safety-critical applications. Our method makes use of certain assumptions, discussed in Section 6, which if violated will lead to guarantees that may no longer hold. In particular, it is important to ensure proper monitoring to detect distribution shift between calibration and deployment.

## Acknowledgements

The authors would like to thank the anonymous reviewers for helpful comments and Hamed Hassani for helpful discussions about the posterior quantile distribution. We also

would like to thank Drew Prinster for bringing several related works to our attention. This work was supported by grant N00014-23-1-2510 from the Office of Naval Research.

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

# A. Theoretical Preliminaries

## A.1. Review of Problem Setup

We first review some relevant aspects of our problem setup.

**Loss Function.** We assume an upper bound on the losses: $\ell_i \in (-\infty, B]$ for $i = 1, \ldots, n$. We assume the same upper bound for $\ell_{\text{new}}$.

**Bayesian Quadrature of Quantile Functions.** Recall Bayes rule for quantile functions:

$$p(K \mid t_{1:n}, \ell_{1:n}) \propto \pi(K) \prod_{i=1}^{n} \delta(\ell_i - K(t_i)), \tag{35}$$

where $\delta$ is the Dirac delta function. The prior $\pi(K)$ is assumed to be sufficiently expressive to have nonzero measure for the set $\mathcal{K}_n$ of quantile functions such that $K(t_i) = \ell_i$ for $i = 1, \ldots, n$ and $K \in \mathcal{K}_n$. This is necessary to prevent the posterior distribution in (35) from becoming degenerate.

## A.2. Background

We begin by recalling some basic properties of distribution functions and quantile functions.

**Proposition A.1** (Properties of Distribution Functions (Shao, 2003, p. 4)). *Let $F(x) = \Pr(X \leq x)$ be a distribution function. Then $F(-\infty) = \lim_{x \to -\infty} F(x) = 0$, $F(\infty) = \lim_{x \to \infty} F(x) = 1$, $F$ is nondecreasing (i.e., $F(x) \leq F(y)$ if $x \leq y$), and $F$ is right continuous (i.e., $\lim_{y \to x, y > x} F(y) = F(x)$).*

Let $F$ be a distribution function and $K(t) \equiv F^{-1}(t) = \inf\{x : F(x) \geq t\}$ be the corresponding quantile function.

**Proposition A.2** (Quantile Functions are Nondecreasing). *If $t \leq u$, then $K(t) \leq K(u)$.*

*Proof.* Since $u \geq t$, it follows that $\{x : F(x) \geq u\} \subseteq \{x : F(x) \geq t\}$. Taking the infimum of both sides yields

$$\inf\{x : F(x) \geq u\} \geq \inf\{x : F(x) \geq t\} \Rightarrow K(u) \geq K(t). \tag{36}$$

$\square$

We also will make use of the probability integral transformation, which we state here for convenience.

**Proposition A.3** (Probability Integral Transformation (Shorack & Wellner, 2009, p. 5)). *If $X$ has distribution function $F$, then*

$$\Pr(F(X) \leq t) \leq t \qquad \text{for all } 0 \leq t \leq 1, \tag{37}$$

*with equality failing if and only if $t$ is not in the closure of the range of $F$. Thus if $F$ is continuous, then $T = F(X)$ is* Uniform$(0, 1)$.

# B. Proof of Results from the Main Paper

## B.1. Proof of Proposition 3.1

Recall that $L_{\text{scp}}(\theta, \lambda)$ is the *miscoverage loss*:

$$\begin{aligned} L_{\text{scp}}(\theta, \lambda) &= \Pr\{s(z_{\text{new}}) > \lambda\} \\ &= 1 - \Pr\{s(z_{\text{new}}) \leq \lambda\} \\ &= 1 - \int \mathbb{1}\{s(z_{\text{new}}) \leq \lambda\} f(z_{\text{new}} \mid \theta) \, dz_{\text{new}}, \end{aligned} \tag{38}$$

where $s$ is an arbitrary nonconformity function.

**Proposition 3.1.** *Define* $s_i \triangleq s(z_i)$ *for* $i = 1, \ldots, n$ *and let* $s_{(1)} \leq s_{(2)} \leq \ldots \leq s_{(n)}$ *be the corresponding order statistics. Let* $\lambda_{\mathrm{scp}}$ *be the following decision rule:*

$$\lambda_{\mathrm{scp}} = \begin{cases} s_{(\lceil (n+1)(1-\alpha) \rceil)}, & \text{if } \lceil (n+1)(1-\alpha) \rceil \leq n \\ \infty, & \text{otherwise.} \end{cases} \tag{12}$$

*Then* $\lambda_{\mathrm{scp}}$ *is an* $\alpha$-acceptable decision rule for the miscoverage loss $L_{\mathrm{scp}}$ defined in (11).

*Proof.* By Lei et al. (2018, Section 2),

$$\Pr(s_{\mathrm{new}} \leq \hat{q}_{1-\alpha}) \geq 1 - \alpha, \tag{39}$$

where

$$\hat{q}_{1-\alpha} = \begin{cases} s_{(\lceil (n+1)(1-\alpha) \rceil)} & \text{if } \lceil (n+1)(1-\alpha) \rceil \leq n \\ \infty, & \text{otherwise.} \end{cases} \tag{40}$$

$\square$

But $L_{\mathrm{scp}}(\theta, \lambda) = 1 - \Pr(s_{\mathrm{new}} \leq \lambda \mid \theta)$, so for $\lambda = \hat{q}_{1-\alpha}$, $R(\theta, \lambda_{\mathrm{scp}}) \leq \alpha$. This statement not depend on $\theta$, and so $\bar{R}(\lambda_{\mathrm{scp}}) \leq \alpha$.

### B.2. Proof of Proposition 3.2

Recall that the $L_{\mathrm{crc}}$ is defined as:

$$L_{\mathrm{crc}}(\theta, \lambda) = \int \ell(z_{\mathrm{new}}, \lambda) f(z_{\mathrm{new}} \mid \theta) \, dz_{\mathrm{new}}, \tag{41}$$

where $\ell(z_{\mathrm{new}}, \lambda)$ is an individual loss function that is monotonically non-increasing in $\lambda$.

**Proposition 3.2.** *Let* $\lambda_{\mathrm{crc}}$ *be the following decision rule:*

$$\lambda_{\mathrm{crc}} = \inf \left\{ \lambda : \frac{1}{n+1} \left( \sum_{i=1}^{n} \ell(z_i, \lambda) + B \right) \leq \alpha \right\}. \tag{15}$$

*Then* $\lambda_{\mathrm{crc}}$ *is an* $\alpha$-acceptable decision rule for $L_{\mathrm{crc}}$ defined in (14).

*Proof.* Let $L_1, \ldots, L_n, L_{n+1}$ be an exchangeable collection of non-increasing random functions $L_i : \Lambda \to (-\infty, B]$. By Angelopoulos et al. (2024, Theorem 1),

$$\mathbb{E}[L_{n+1}(\hat{\lambda})] \leq \alpha, \tag{42}$$

where

$$\hat{\lambda} = \inf \left\{ \lambda : \frac{n}{n+1} \hat{R}_n(\lambda) + \frac{B}{n+1} \leq \alpha \right\} \tag{43}$$

and $\hat{R}_n(\lambda) = (L_1(\lambda) + \ldots + L_n(\lambda))/n$.

Interpreting these results using the notation from Section 3 of the main paper, we identify:

- $L_i(\lambda) = \ell(z_i, \lambda)$ for $i = 1, \ldots, n$ and $L_{n+1}(\lambda) = \ell(z_{\mathrm{new}}, \lambda)$,

- $\lambda_{\mathrm{crc}}$ is identical to $\hat{\lambda}$ from (43), and

- (42) states that $R(\theta, \lambda_{\mathrm{crc}}) \leq \alpha$ for any $\theta$.

Therefore, $\bar{R}(\lambda_{\mathrm{crc}}) = \sup_\theta R(\theta, \lambda_{\mathrm{crc}}) \leq \alpha$. $\square$

## B.3. Proof of Theorem 4.1

In order to prove Theorem 4.1, we will need to make use of two auxiliary propositions (Proposition B.1 and Proposition B.2). We state and prove these first, and then proceed to prove Theorem 4.1.

**Proposition B.1.** *Consider the following variational maximization problem:*

$$I[f] = \int_a^b f(x)\, dx \tag{44}$$

*subject to $f(a) = f_a$, $f(b) = f_b$, and $f_a \leq f(x) \leq f_b$ for all $x \in [a, b]$, where $f_a \leq f_b$. Then $I[f]$ is maximized by*

$$f^*(x) = \begin{cases} f_a & \text{if } x = a, \\ f_b & \text{otherwise,} \end{cases} \tag{45}$$

*and $I[f^*] = (b - a)f_b$.*

*Proof.* We apply Euler's method (Kot, 2014, Section 2.2), which approximates the variational problem as an $m$-dimensional problem and takes the limit as $m \to \infty$. Let the interval $[a, b]$ be divided into $m+1$ subintervals of equal width $\Delta x = \dfrac{b - a}{m + 1}$. The objective functional can then be approximated as

$$I(f_1, \ldots, f_m) \equiv \sum_{j=0}^{m} f_j \Delta x, \tag{46}$$

where $f_0 = f_a$ and $f_{m+1} = f_b$ due to the boundary conditions. In order to handle the $f_a \leq f(x) \leq f_b$ constraint, we first impose $f(x) \leq f_b$ and check if the solution also satisfies $f(x) \geq f_a$. To that end, we substitute $f_j = f_b - \xi_j^2$:

$$I(\xi_1, \ldots, \xi_m) = \sum_{j=0}^{m} (f_b - \xi_j^2)\Delta x. \tag{47}$$

We then take partial derivatives with respect to $\xi_k$:

$$\frac{\partial I}{\partial \xi_k} = -2\xi_k \Delta x \Rightarrow \frac{1}{\Delta x}\frac{\partial I}{\partial \xi_k} = -2\xi_k. \tag{48}$$

Taking the limit as $m \to \infty$ and $\Delta x \to 0$, the variational derivative becomes:

$$\frac{\delta I}{\delta \xi} = -2\xi. \tag{49}$$

Setting $\dfrac{\delta I}{\delta \xi} = 0$ yields $\xi(x) = 0$, which recovers $f(x) = f_b$, except at $x = a$, where $f(a) = f_a$ by the boundary conditions. This recovers $f^*(x)$ from (45), which indeed satisfies $f(x) \geq f_a$. For $f^*$, it is evident that the value of the functional is $I[f^*] = (b - a)f_b$. □

**Proposition B.2.** *Let $\mathcal{K}_n$ be the set of quantile functions for which $K(t_i) = \ell_i$ for $i = 1, \ldots, n$. Then*

$$\sup_{K \in \mathcal{K}_n} J[K] = \sum_{i=1}^{n+1} (t_{(i)} - t_{(i-1)})\ell_{(i)}, \tag{50}$$

*where $t_{(0)} = 0$, $t_{(n+1)} = 1$, $\ell_{(n+1)} = B$, and $J[K] \triangleq \int_0^1 K(t)\, dt$.*

*Proof.* By Proposition A.2, quantile functions preserve orderings and therefore $K(t_{(i)}) = \ell_{(i)}$. We divide $J[K]$ into intervals with endpoints $(0, t_{(1)}), (t_{(1)}, t_{(2)}), \ldots, (t_{(n)}, 1)$:

$$\sup_{K \in \mathcal{K}_n} J[K] = \sup_{K \in \mathcal{K}_n} \int_0^1 K(t)\, dt \tag{51}$$

$$= \sup_{K \in \mathcal{K}_n} \sum_{i=1}^{n+1} \int_{t_{(i-1)}}^{t_{(i)}} K(t)\, dt \tag{52}$$

$$\leq \sum_{i=1}^{n+1} \sup_{K \in \mathcal{K}_n} \int_{t_{(i-1)}}^{t_{(i)}} K(t)\, dt \tag{53}$$

By Proposition A.2, $K(t_{(i-1)}) \leq K(t) \leq K(t_{(i)})$ for any $t \in [t_{(i-1)}, t_{(i)}]$. We view each term as a variational subproblem where $J_i[K_i] \triangleq \int_{t_{(i-1)}}^{t_{(i)}} K_i(t)\, dt$ with boundary conditions $K_i(t_{(i-1)}) = \ell_{(i-1)}$ and $K_i(t_{(i)}) = \ell_{(i)}$. We therefore appeal to Proposition B.1 to conclude that

$$K_i^*(t) = \begin{cases} \ell_{(i-1)} & \text{if } t = t_{(i-1)}, \\ \ell_{(i)} & \text{otherwise,} \end{cases} \tag{54}$$

and $J[K_i^*] = (t_{(i)} - t_{(i-1)})\ell_{(i)}$. We therefore have

$$\sup_{K \in \mathcal{K}_n} J[K] \leq \sum_{i=1}^{n+1} (t_{(i)} - t_{(i-1)})\ell_{(i)}. \tag{55}$$

By composing $K_i^*$ from each subinterval, it is straightforward to see that the bound is tight for

$$K_{t_{1:n}, \ell_{1:n}}^*(t) = \begin{cases} \ell_{(1)} & \text{if } t \leq t_{(1)} \\ \ell_{(2)} & \text{if } t_{(1)} < t \leq t_{(2)} \\ \ldots \\ \ell_{(n)} & \text{if } t_{(n-1)} < t \leq t_{(n)} \\ B & \text{if } t > t_{(n)}. \end{cases} \tag{56}$$

$K_{t_{1:n}, \ell_{1:n}}^*$ is therefore the "worst-case" quantile function that is consistent with the observations, and $J[K_{t_{1:n}, \ell_{1:n}}^*] = \sum_{i=1}^{n+1} (t_{(i)} - t_{(i-1)})\ell_{(i)}$. $\qquad\square$

We are now ready to prove Theorem 4.1.

**Theorem 4.1.** *Let $t_{(0)} = 0$, $t_{(n+1)} = 1$, and $\ell_{(n+1)} = B$. Then*

$$\sup_\pi E(L \mid t_{1:n}, \ell_{1:n}) \leq \sum_{i=1}^{n+1} u_i \ell_{(i)}, \tag{26}$$

*where $u_i = t_{(i)} - t_{(i-1)}$.*

*Proof.* Let $J[K] = \int_0^1 K(t)\, dt$. The conditional expected loss can be expressed as:

$$E(L \mid t_{1:n}, \ell_{1:n}) = \int J[K] p(K \mid t_{1:n}, \ell_{1:n})\, dK \tag{57}$$

$$\leq \sup_{K \in \mathcal{K}_n} J[K], \tag{58}$$

where $\mathcal{K}_n$ is the set of quantile functions for which $K(t_i) = \ell_i$ for $i = 1, \ldots, n$. By Proposition B.2, it follows that

$$E(L \mid t_{1:n}, \ell_{1:n}) \leq \sum_{i=1}^{n+1} (t_{(i)} - t_{(i-1)})\ell_{(i)} = \sum_{i=1}^{n+1} u_i \ell_{(i)} \tag{59}$$

$$\square$$

## B.4. Proof of Lemma 4.2

**Lemma 4.2** (Distribution of Quantile Spacings (Aitchison & Dunsmore, 1975, p. 140)). *Suppose that $\ell_1, \ldots, \ell_n$ are drawn i.i.d. with continuous[5] distribution function $F$. Let $t_i = F(\ell_i)$ and $u_i = t_{(i)} - t_{(i-1)}$, where by convention $t_{(0)} = 0$ and $t_{(n+1)} = 1$. Then $(u_1, u_2, \ldots, u_{n+1}) \cong \mathrm{Dir}(1, \ldots, 1)$.*

*Proof.* By the probability integral transformation (Proposition A.3), $T_i$ is $\mathrm{Uniform}(0,1)$ for $i = 1, \ldots, n$. Since the transformation from $(t_1, \ldots, t_n) \to (t_{(1)}, \ldots, t_{(n)})$ is a sorting operation where $n!$ permutations map to the same vector of order statistics, the probability density for $t_{(1)}, \ldots, t_{(n)}$ is therefore

$$f_{t_{(1:n)}}(t_{(1)}, \ldots, t_{(n)}) = n!, \qquad 0 \leq t_{(1)} \leq t_{(2)} \leq \ldots \leq t_{(n)} \leq 1. \tag{60}$$

If $u_{1:n} = G(t_{(1:n)})$ where $G$ is differentiable and invertible, then by change of variables the density for $u_{1:n}$ can be expressed as

$$f_{u_{1:n}}(u_{1:n}) = f_{t_{(1:n)}}(G^{-1}(u_{1:n})) \left| \det\left( \frac{\partial}{\partial u_{1:n}} G^{-1}(u_{1:n}) \right) \right|. \tag{61}$$

Observe that the inverse transformation $t_{(1:n)} = G^{-1}(u_{1:n})$ can be expressed as

$$\begin{bmatrix} t_{(1)} \\ t_{(2)} \\ t_{(3)} \\ \vdots \\ t_{(n-1)} \\ t_{(n)} \end{bmatrix} = \begin{bmatrix} 1 & 0 & 0 & \ldots & 0 & 0 \\ 1 & 1 & 0 & \ldots & 0 & 0 \\ 1 & 1 & 1 & \ldots & 0 & 0 \\ \vdots & \vdots & \vdots & \ddots & \vdots & \vdots \\ 1 & 1 & 1 & \ldots & 1 & 0 \\ 1 & 1 & 1 & \ldots & 1 & 1 \end{bmatrix} \begin{bmatrix} u_1 \\ u_2 \\ u_3 \\ \vdots \\ u_{n-1} \\ u_n \end{bmatrix}. \tag{62}$$

Hence the absolute Jacobian of inverse transformation $t_{(1:n)} = G^{-1}(u_{1:n})$ is 1. The density of $u_{1:n}$ is therefore

$$f_{u_{1:n}}(u_{1:n}) = f_{t_{(1:n)}}(G^{-1}(u_{1:n})) = n!, \quad \text{where } u_i \geq 0 \text{ for } i = 1, \ldots, n \text{ and } \sum_{i=1}^{n} u_i \leq 1. \tag{63}$$

Recall that the Dirichlet density with parameter $\alpha_1, \ldots, \alpha_{n+1}$ is:

$$\mathrm{Dir}(u_{1:n+1} \mid \alpha_{1:n+1}) = \frac{\Gamma(\sum_{i=1}^{n+1} \alpha_i)}{\Gamma(\alpha_1) \ldots \Gamma(\alpha_{n+1})} \prod_{i=1}^{n+1} u_i^{\alpha_i - 1}, \quad \text{where } u_i \geq 0 \text{ and } \sum_{i=1}^{n+1} u_i = 1. \tag{64}$$

In particular, if $\alpha_1 = \alpha_2 = \ldots = \alpha_{n+1} = 1$,

$$\mathrm{Dir}(u_{1:n+1} \mid 1, \ldots, 1) = \Gamma(n+1) = n!, \tag{65}$$

which is identical to (63) with $u_{n+1} = 1 - u_1 - \ldots - u_n$. Therefore, $(u_1, u_2, \ldots, u_{n+1}) \cong \mathrm{Dir}(1, \ldots, 1)$.

$\square$

## B.5. Proof of Theorem 4.3

**Theorem 4.3.** *Define $\ell_{(i)}$ to be the order statistics of $\ell_1, \ldots, \ell_n$ for $i = 1, \ldots, n$ and $\ell_{(n+1)} \triangleq B$. Let $L^+$ be the random variable defined as follows:*

$$U_1, \ldots, U_{n+1} \sim \mathrm{Dir}(1, \ldots, 1), \quad L^+ = \sum_{i=1}^{n+1} U_i \ell_{(i)}. \tag{27}$$

*Then for any $b \in (-\infty, B]$,*

$$\inf_{\pi} \Pr(L \leq b \mid \ell_{1:n}) \geq \Pr(L^+ \leq b). \tag{28}$$

---

[5]The correspondence to a Dirichlet distribution holds exactly for continuous distributions. Weighted sums of Dirichlet random variates stochastically dominate weighted sums of discrete quantile spacings, and thus due to space constraints we only consider continuous distributions here.

*Proof.*

$$\inf_{\pi} \Pr(L \leq b \mid \ell_{1:n}) = \inf_{\pi} \int \mathbb{1}\left\{J[K] \leq b\right\} p(K \mid \ell_{1:n}) \, dK \tag{66}$$

$$= \inf_{\pi} \int \mathbb{1}\left\{J[K] \leq b\right\} \left( \int p(K \mid t_{1:n}, \ell_{1:n}) p(t_{1:n} \mid \ell_{1:n}) \, dt_{1:n} \right) dK \tag{67}$$

$$= \inf_{\pi} \int \left( \int \mathbb{1}\left\{J[K] \leq b\right\} p(K \mid t_{1:n}, \ell_{1:n}) \, dK \right) p(t_{1:n} \mid \ell_{1:n}) \, dt_{1:n} \tag{68}$$

$$\geq \int \left( \inf_{\pi} \int \mathbb{1}\left\{J[K] \leq b\right\} p(K \mid t_{1:n}, \ell_{1:n}) \, dK \right) p(t_{1:n} \mid \ell_{1:n}) \, dt_{1:n} \tag{69}$$

$$\geq \int \left( \inf_{K \in \mathcal{K}_n} \mathbb{1}\left\{J[K] \leq b\right\} \right) p(t_{1:n} \mid \ell_{1:n}) \, dt_{1:n} \tag{70}$$

$$\geq \int \mathbb{1}\left\{J[K^*_{t_{1:n}, \ell_{1:n}}] \leq b\right\} p(t_{1:n} \mid \ell_{1:n}) \, dt_{1:n} \tag{71}$$

$$= \int \mathbb{1}\left\{\sum_{i=1}^{n+1}(t_{(i)} - t_{(i-1)})\ell_{(i)} \leq b\right\} p(t_{1:n} \mid \ell_{1:n}) \, dt_{1:n} \tag{72}$$

$$= \int \mathbb{1}\left\{\sum_{i=1}^{n+1} u_i \ell_{(i)} \leq b\right\} p(u_{1:n+1} \mid \ell_{1:n}) \, du_{1:n+1} \tag{73}$$

$$= \Pr(L^+ \leq b) \tag{74}$$

$\square$

Note that here the quantile distribution $p(t_{1:n} \mid \ell_{1:n})$ takes into account the ordering of the losses but not the values themselves. This is consistent with the implicit assumption, made in the frequentist treatment of conformal prediction, that a new observation is equally likely to fall into each of the intervals formed by the loss values and the endpoints. The results for characterizing this distribution can be traced back to Tukey (1947), by way of Guttman (1970) and Aitchison & Dunsmore (1975), and is also known as $A_n$ in the literature (Hill, 1968; 1988). These results allow us to make minimal assumptions about the form of the CDF.

### B.6. Proof of Corollary 4.4

**Corollary 4.4.** *For any desired confidence level $\beta \in (0, 1)$, define*

$$b^*_\beta = \inf_b \{b : \Pr(L^+ \leq b \mid \ell_{1:n}) \geq \beta\}. \tag{29}$$

*Then* $\inf_\pi \Pr(L \leq b \mid \ell_{1:n}) \geq \beta$ *for any* $b \geq b^*_\beta$.

*Proof.* For any $b \geq b^*_\beta$, $\Pr(L^+ \leq b \mid \ell_{1:n}) \geq \beta$. Substitution into (28) provides the desired result. $\square$

