# OpenReview forum: "Conformal Prediction as Bayesian Quadrature"
_ICML.cc/2025/Conference — ICML 2025 oral_

### Official Review · Reviewer_dgCV · 2025-02-23

**Overall Recommendation:** 4

**Summary:**

This paper proposes a new Bayesian interpretation of conformal prediction, which recovers standard conformal prediction as its mean and provides additional finite sample uncertainty estimate. Interestingly, as a Bayesian algorithm, the uncertainty comes from both 'lack of precise input locations' and finite (insufficient) number of observations. Another nice thing is that by taking the supremum of all possible priors, the practitioners do not need to worry about the choice of prior, which is usually a huge headache in practice. In the experiments, the advantage of the proposed method is clearly demonstrated.

**Claims And Evidence:**

Yes, the paper is well written.

**Essential References Not Discussed:**

Line 376-377. This paper claims that the Bayesian interpretation allows conditional coverage guarantees. I suggest the authors distingush their contributions from the real conditional coverage guarantee, find $\hat{C}\left(X_{n+1}\right)$ such that $\mathbb{P}\left(Y_{n+1} \in \hat{C}\left(X_{n+1}\right) \mid X_{n+1}=x\right)=1-\alpha$. See for example, https://arxiv.org/pdf/2305.12616, https://arxiv.org/abs/1903.04684. From my understanding, the authors' Bayesian interpretation can still only provide marginal coverage guarantees. I request the authors to make this distinction in the paper.

**Experimental Designs Or Analyses:**

Why the authors only include conformal risk control in the experiments and not do anything about split conformal prediction, given that the method works for both settings. I am quite familar with split conformal prediction, but I have to admit I know little about conformal risk control. I am not asking for more experiments, I am curious what make the authors only include experiments on conformal risk control.

**Methods And Evaluation Criteria:**

N/A

**Other Comments Or Suggestions:**

N/A

**Other Strengths And Weaknesses:**

Strength:
1. The section 4 makes serveal very interesting observations. 1) The standard expectation can be translated to an integral of the inverse CDF function. Then, even if the input locations are not known, one can deduce that they follow a Dirichlet distribution from a classical result presented as Lemma 4.2. As a result, the dirichilet distribution over the input location quantifies an upper bound on the CDF of the final integral. 2) The posterior mean can be upper bounded by the integral of the 'worst-case' quantile function that is consistent with the observations. I really enjoy reading this section and read the proofs in the appendix. It is very enlightening.

Weakness:
1. The way Bayesian quadrature is written is not standard in the BQ literature as I am aware of. The authors use the standard notation for regression setting, i.e Equation (15) in https://arxiv.org/pdf/1807.02582, while personally I prefer more the interpolation notation for Bayesian quadrature, see Equation (20) in https://arxiv.org/pdf/1807.02582, as the likelihood function is degenerate.

**Questions For Authors:**

N/A

**Relation To Broader Scientific Literature:**

N/A

**Theoretical Claims:**

I have checked all the proofs in the appendix. The proofs are well-written and easy to follow.

---

> ### Author Rebuttal · Authors · 2025-04-01
>
> We thank the reviewer for writing a thoughtful and detailed review. We are happy that the review has recognized that the choice of prior can be a headache in practical problems and how our method circumvents this. We also thank the reviewer for reading our proofs. We are glad that the reviewer found them to be enlightening.
>
> **Conformal Risk Control vs. Split Conformal Prediction.** We thank the reviewer for raising the point about the experiments being run on Conformal Risk Control rather than split conformal prediction. The main reason that the experiments focus on Conformal Risk Control is pragmatic in nature. CRC recovers split conformal prediction when the miscoverage loss is used (i.e. loss = 1 if prediction set/interval does not cover ground truth and loss = 0 otherwise), but allows more general loss functions to be considered. Therefore, in order to investigate more complex loss functions, the experiments focus on CRC. However, please see our next point below which includes results on Split Conformal Prediction.
>
> **Additional Experiments.** During the rebuttal period we ran additional experiments in a more traditional prediction interval setting with heteroskedastic data. In each of the 10,000 random trials we use 200 calibration samples. We let $X \sim U[0, 4]$ and $Y|X \sim \mathcal{N}(0, X^2)$. Prediction intervals are formed as $[-\hat{\lambda}, \hat{\lambda}]$ where $\hat{\lambda}$ is to be selected by each method. The loss is the miscoverage loss and the target loss is set to 0.1 (i.e. 90% coverage). The maximum allowable risk failure rate is set to 5% (i.e. $\beta = 0.95$). Please note that since the miscoverage loss is used here, Conformal Risk Control and Split Conformal Prediction coincide.
>
> | Method | Relative Freq. (Failure Rate) | 95% CI | Mean Prediction Interval Length |
> |---|---|---|---|
> | Split Conformal Prediction / CRC | 46.59% | [45.61%, 47.57%] | 7.67 |
> | RCPS | 0.0% | [0.0%, 0.04%] | 13.99 |
> | Ours ($\beta = 0.95$) | 3.75% | [3.39%, 4.14%] | 9.14 |
>
> The results indicate that even in the heteroskedastic setting, our method allow more precise control of $\hat{\lambda}$. Unlike RCPS, which is overly conservative, ours produces the shortest prediction intervals while not violating the maximum allowable risk failure rate.
>
> **On the nature of conditional coverage.** We thank the reviewer for raising this point. Previous work on conditional guarantees have focused on input-conditional guarantees, where the guarantee is conditioned on $X_{n+1} = x$ for all $x$ in the input domain. Guarantees of this nature have been shown to be generally impossible without stronger distribution assumptions. Our guarantees are perhaps better characterized by the term "data-conditional guarantee", where we condition on the set of observed loss values $\ell_{1:n}$. Our experiments demonstrate the practical benefits of this by achieving decisions that produce smaller prediction sets and intervals while not violating the constraint on maximum allowable failure rate. Our guarantees do not rely on strong distribution assumptions that would be necessary to produce an input-conditional guarantee. The distinction between input-conditional and our data-conditional guarantees is an important one and we will be sure to clarify this in the paper by adding a paragraph discussing this point.
>
> **Notation.** We thank the reviewer for suggesting improvements to the notation. We would be happy to update the notation for clarity. However, unfortunately, the links in the review seem to be pointing to the same paper. If the reviewer would be so kind as to re-post the links in a comment, we would be glad to update our paper accordingly.

---

> > ### Comment · Reviewer_dgCV · 2025-04-02
> >
> > Thank the authors for their rebuttals. I do not have further questions.
> >
> > I prefer more the interpolation notation for Bayesian quadrature, see Equation (20) in https://arxiv.org/pdf/1807.02582, as the likelihood function is degenerate.

---

### Official Review · Reviewer_gGKC · 2025-03-08

**Overall Recommendation:** 4

**Summary:**

The paper proposes a Bayesian quadrature approach as a Bayesian alternative to conformal prediction, encompassing two widely used methods: split conformal prediction and conformal risk control. The equivalence between these approaches and Bayesian quadrature is clearly established through theoretical proofs. Empirically, the proposed Bayesian method demonstrates strong performance compared to frequentist alternatives, particularly in terms of lower risk violation frequency and more compact prediction sets.

## Update after rebuttal
I found the authors’ rebuttal satisfactory. I have maintained my score 4.

**Claims And Evidence:**

The paper claims that the proposed Bayesian quadrature approach generalizes conformal prediction. Specifically, Propositions 3.1 and 3.2 demonstrate that split conformal prediction and conformal risk control can be reinterpreted within a decision-theoretic framework, while equations (30)–(32) show that the expected loss of the quantile spacing approach recovers the corresponding conformal methods. These claims are supported by theoretical proofs.
However, given my limited knowledge of conformal prediction, I am unsure whether these two methods alone are sufficient to establish that Bayesian quadrature is a generalization of conformal prediction. In this sense, the title may be somewhat strong. That said, I do agree that the proposed approach encompasses these two conformal methods.

**Essential References Not Discussed:**

The literature review on Bayesian quadrature is relatively limited to fundamental works, which makes sense given that this paper’s approach differs significantly from recent advancements in Bayesian quadrature that rely on functional priors, such as Gaussian processes and kernel methods. Therefore, I have no concerns regarding the references.

**Experimental Designs Or Analyses:**

I reviewed the experimental results but did not examine all the details, such as the implementation code. However, the findings appear reasonable to me.

**Methods And Evaluation Criteria:**

They first establish the mathematical equivalence between their Bayesian quadrature approach and existing conformal prediction methods, specifically split conformal prediction and conformal risk control. They then present experimental results demonstrating that their proposed Bayesian posterior risk approach reduces the number of individual trials that exceed the target risk threshold while maintaining a smaller prediction set size compared to existing conformal methods.

**Other Comments Or Suggestions:**

I did not find any typos.

**Other Strengths And Weaknesses:**

Strengths:
1. I like the novel attempt to leverage Bayesian quadrature to reformulate conformal prediction. This perspective is interesting and bridges two distinct fields.
2. Applying a Bayesian approach to random quantile spacings is a clever idea, particularly in the context of integrating monotonic integrand functions like quantiles and cumulative density functions. Traditional kernel-based methods often struggle to enforce monotonicity from a functional prior, typically requiring crude approximations such as warped Gaussian processes. This method provides an interesting alternative.
3. The results appear promising. The improved risk violation and sample efficiency suggest a compelling direction for further research.

Weakness:
1. The Bayesian interpretation is somewhat unclear. I suspect this is because the paper is primarily written for the conformal prediction community, but some parts give the impression of denying Bayesian principles. For instance, Section 4.3 mentions the "elimination of the prior distribution," which, strictly speaking, contradicts the Bayesian viewpoint, where priors are valuable for incorporating contextual information beyond the observed data. I would appreciate a clearer explanation of the role of the prior in this approach. If the authors are referring to an uninformative prior, a prior that is robust against variation, or a weighted discrete distribution as the prior, it would be helpful to state this explicitly.
2. Figure 1 could be misleading. It depicts the typical Bayesian quadrature procedure, where a functional prior is placed over the quantile function $F^{-1}$, assuming smooth monotonicity. However, if I understand correctly, the authors do not ultimately place a function space prior directly. Instead, they propose placing a Dirichlet prior on quantile spacings and using a Heaviside step function (akin to an empirical CDF) as a deterministic function, then this forms the non-parametric functional prior. The resulting posterior over varying spacings then induces a smooth posterior over the integral in the expected values of $L^{+}$. Is this interpretation correct? Clarifying this aspect would improve the paper’s clarity.

**Questions For Authors:**

See weakness section.

**Relation To Broader Scientific Literature:**

Quantifying the uncertainty of black-box predictors is crucial for high-stakes applications and exploratory tasks. I expect this paper to benefit various decision-making domains, including autonomous driving, experimental design, and time-series forecasting.

**Theoretical Claims:**

I have not verified all the theoretical proofs in detail, but the claims appear reasonable to me.

---

> ### Author Rebuttal · Authors · 2025-04-01
>
> We thank the reviewer for writing a detailed review that recognizes the bridging nature of our work and its benefits for various decision-making domains.
>
> **Generalizing Conformal Prediction.** One of our main contributions is to show how both split conformal prediction and Conformal Risk Control can be recovered by taking the posterior mean of our upper bounding loss random variable $L^+$. Thus, our method illuminates a broader viewpoint and uses this insight to choose $\lambda$ more effectively. We believe this is generally useful perspective that may have broader implications on conformal prediction in general. Nevertheless, we will update the text of the manuscript to clarify the nature of the generalization elaborated here (i.e. oriented towards split conformal prediction and Conformal Risk Control).
>
> **Role of the prior.** We thank the reviewer for raising this nuanced but important point. Our goal is to show that the Bayesian viewpoint unlocks a richer interpretation compared to previous works, which focus on marginal guarantees that as we have shown in the paper correspond to the posterior mean. For better or for worse, there is still a big gap in the literature between traditional approaches to distribution-free uncertainty quantification, which are predominantly frequentist in nature, and methods like Bayesian quadrature which are firmly Bayesian in nature. Therefore, to draw an explicit correspondence between the two, the dependence on the prior is removed in Section 4.3. The intuition is that that any rational decision maker operating according to the rules of probability, regardless of prior (sufficiently expressive), would agree with the upper-bounding distribution of $L^+$ we derive. Naturally, commitment to a specific choice of prior would lead to tighter distributions over the posterior risk, and in future work we seek to bridge these fields even further by exploring specific choices of priors over quantile functions. We will add a paragraph explicitly discussing these points to the manuscript.
>
> **Figure 1.** Thank you for raising this point. The interpretation stated in the review is correct: the smoothness does indeed result from the combination of a step function with the distribution over quantile spacings. Our original intention was to illustrate the general idea of Bayesian quadrature in our setting with Figure 1, and then move into the specifics of our method in Figure 2. However, we will update Figure 1 to more clearly signpost the use of the step function, both in the caption and by updating the figure itself.

---

### Official Review · Reviewer_vkyV · 2025-03-12

**Overall Recommendation:** 5

**Summary:**

The authors propose a Bayesian version of conformal prediction, which guarantees conditional coverage, rather than marginal coverage.  The technique is distribution free, since it considers the worst case risk by maximizing over all possible priors. This is made tractable by leveraging some prior results on distribution free analysis of quantile spacings, combined with Bayesian quadrature. Their method includes frequentist split conformal prediction and conformal risk control (CRC)  as special cases.They show significantly improved results (in terms of risk control and prediction set size) on synthetic data, and on controlling the false negative rate of multilabel classification on the MSCOCO, compared to  CRC and Risk-controlling Prediction Sets (RCPS). Overall a very impressive paper.

**Claims And Evidence:**

Theoretical claims (see summary) are supported by proofs (not checked) and compelling experimental results.

**Essential References Not Discussed:**

NA

**Experimental Designs Or Analyses:**

Experiments seem sound.

**Methods And Evaluation Criteria:**

Yes, good eval on synthetic data and a standard challenging real world image classification benchmark.

**Other Comments Or Suggestions:**

Emphasize that your method is conditional on observed data, and is better than a marginal guarantee.
This part of Bayes is more important than using a prior (which you avoid).

**Other Strengths And Weaknesses:**

As I said above, extremely strong paper.

**Questions For Authors:**

NA

**Relation To Broader Scientific Literature:**

Related work is very well explained. This particular combination of techniques (Bayesian quadrature and distribution-free analysis of random quantil spacings) seems entirely novel, and is very creative.

**Theoretical Claims:**

I did not check the correctness of the proofs.

---

> ### Author Rebuttal · Authors · 2025-04-01
>
> We thank the reviewer for taking the time to write a thoughtful and detailed review. We are pleased that the review recognized the novelty of creatively combining Bayesian quadrature and distribution-free analysis of random quantile spacings.
>
> We will update the paper to clarify that our guarantees are conditional on the observed calibration data. In essence, our guarantees are probabilistic statments about the risk conditioned on the observed losses (see e.g. the conditioning on $\ell_{1:n}$ in Theorem 4.3 and Corollary 4.4). This stands in contrast to previous approaches which rely on marginalizing over many possible realizations of the calibration losses that were not observed. We will add a paragraph to the paper explicitly clarifying this point.

---

### Official Review · Reviewer_K6B1 · 2025-03-15

**Overall Recommendation:** 3

**Summary:**

This paper proposes a Bayesian reinterpretation of conformal prediction, framing it within a Bayesian quadrature framework. The authors shows that split conformal prediction and conformal risk control can be derived as special cases of Bayesian quadrature. By modeling uncertainty over quantile functions and leveraging Dirichlet-distributed quantile spacings, they derive a posterior distribution over expected losses, enabling more interpretable and adaptive risk guarantees.

**Claims And Evidence:**

Yes.

**Essential References Not Discussed:**

n/a

**Experimental Designs Or Analyses:**

The two datasets incorporated are well-designed and illuminating. I would love to see more experiments though, to provide some intuition on the strengths & drawbacks of the Bayesian quadrature method in practice. For example, would it be possible to test on synthetic data with non-monotonic loss functions or heteroscedastic noise?

**Methods And Evaluation Criteria:**

Yes.

**Other Comments Or Suggestions:**

n/a

**Other Strengths And Weaknesses:**

Coming from some one who is mostly familiar with the frequentist side of things, this paper provides very interesting and novel insights about UQ & decision making. It shines light on the conservatism (the method needs to work for *any* prior), and failure modes of CRC because of the limitation of expectation. Although the empirical experiments are not comprehensive, the theoretical contributions and interesting enough to make up for it.

**Questions For Authors:**

See experiments section.

**Relation To Broader Scientific Literature:**

There have been attempts to formulate/use conformal prediction for Bayesian inference ([e.g.](https://arxiv.org/abs/2210.12496)), but non as unified as the proposed method.

**Theoretical Claims:**

As far as I can tell, the theoretical claims are correct.

---

> ### Author Rebuttal · Authors · 2025-04-01
>
> We thank the reviewer for taking the time to write a thoughtful and detailed review of our work. We appreciate that the review has recognized the "very interesting and novel insights about UQ & decision making" provided by our paper.
>
> We appreciate the desire for additional experiments to provide additional insight about our method. To this end, we have implemented experiments on heteroskedastic data. The findings are largely in line with the results from Section 5 of our paper: our Bayesian interpretation produces prediction intervals that are shorter than baselines while not exceeding the maximum acceptable failure rate.
>
> In each of the 10,000 random trials we use 200 calibration samples. To achieve heteroskedasticity, we let $X \sim U[0, 4]$ and $Y|X \sim \mathcal{N}(0, X^2)$. Prediction intervals are formed as $[-\hat{\lambda}, \hat{\lambda}]$ where $\hat{\lambda}$ is to be selected by each method. The loss is the miscoverage loss and the target loss is set to 0.1 (i.e. 90% coverage). The maximum allowable risk failure rate is set to 5% (i.e. $\beta = 0.95$). Please note that since the miscoverage loss is used here, Conformal Risk Control and Split Conformal Prediction coincide.
>
> | Method | Relative Freq. (Failure Rate) | 95% CI | Mean Prediction Interval Length |
> |---|---|---|---|
> | Split Conformal Prediction / CRC | 46.59% | [45.61%, 47.57%] | 7.67 |
> | RCPS | 0.0% | [0.0%, 0.04%] | 13.99 |
> | Ours ($\beta = 0.95$) | 3.75% | [3.39%, 4.14%] | 9.14 |
>
> The results indicate that even in the heteroskedastic setting, our method allow more precise control of $\hat{\lambda}$. Unlike RCPS, which is overly conservative, ours produces the shortest prediction intervals while not violating the maximum allowable risk failure rate.

---

> > ### Comment · Reviewer_K6B1 · 2025-04-02
> >
> > Thank you for the additional experiments! Really cool work.

---

### Decision · Program_Chairs · 2025-05-01

**Decision:**

Accept (oral)

**Comment:**

This paper develops a Bayesian perspective for conformal prediction (CP) using a Bayesian quadrature approach. It derives two widely used CP methods as special cases, namely, split CP and conformal risk control. One of the main benefits of this perspective is that it allows conditional coverage guarantees. The paper has strong theoretical and empirical results to demonstrate its benefits over frequentist baselines on both synthetic and real data.

All the reviewers' are positive about this paper and rightly so! This is a refreshingly novel paper and I recommend accepting this paper with an oral presentation at the conference.